# Critical role of deadenylation in regulating poly(A) rhythms and circadian gene expression

Xiangyu Yao[1,2], Shihoko Kojima[1,3], Jing Chen[1,3]*

**1** Department of Biological Sciences, Virginia Polytechnic Institute and State University, Blacksburg, Virginia, United States of America, **2** Genetics, Bioinformatics, and Computational Biology program, Virginia Polytechnic Institute and State University, Blacksburg, Virginia, United States of America, **3** Fralin Life Sciences Institute, Virginia Polytechnic Institute and State University, Blacksburg, Virginia, United States of America

* chenjing@vt.edu

**Data Availability Statement:** All relevant data are within the manuscript and its Supporting Information files.

**Funding:** This project was supported by the startup fund (to J.C.) from the Department of Biological

## Abstract

The mammalian circadian clock is deeply rooted in rhythmic regulation of gene expression. Rhythmic transcriptional control mediated by the circadian transcription factors is thought to be the main driver of mammalian circadian gene expression. However, mounting evidence has demonstrated the importance of rhythmic post-transcriptional controls, and it remains unclear how the transcriptional and post-transcriptional mechanisms collectively control rhythmic gene expression. In mouse liver, hundreds of genes were found to exhibit rhythmicity in poly(A) tail length, and the poly(A) rhythms are strongly correlated with the protein expression rhythms. To understand the role of rhythmic poly(A) regulation in circadian gene expression, we constructed a parsimonious model that depicts rhythmic control imposed upon basic mRNA expression and poly(A) regulation processes, including transcription, deadenylation, polyadenylation, and degradation. The model results reveal the rhythmicity in deadenylation as the strongest contributor to the rhythmicity in poly(A) tail length and the rhythmicity in the abundance of the mRNA subpopulation with long poly(A) tails (a rough proxy for mRNA translatability). In line with this finding, the model further shows that the experimentally observed distinct peak phases in the expression of deadenylases, regardless of other rhythmic controls, can robustly cluster the rhythmic mRNAs by their peak phases in poly(A) tail length and abundance of the long-tailed subpopulation. This provides a potential mechanism to synchronize the phases of target gene expression regulated by the same deadenylases. Our findings highlight the critical role of rhythmic deadenylation in regulating poly(A) rhythms and circadian gene expression.

## Author summary

The biological circadian clock aligns bodily functions to the day-and-night cycle and is important for maintaining health. The rhythms in various biological processes ultimately stem from rhythmic gene expression in each single cell. Because several proteins in the

Sciences, Virginia Tech. X.Y. was partially supported by a fellowship from the GBCB graduate program at Virginia Tech. Virginia Tech Open Access Subvention Fund (OASF) provided financial support for the publication fee. The funders had no role in study design, data collection and analysis, decision to publish, or preparation of the manuscript.

**Competing interests:** The authors have declared that no competing interests exist.

mammalian core clock machinery are transcription factors, studies of mammalian circadian gene expression have focused on rhythmic transcriptional control. However, many recent studies have suggested the importance of rhythmic post-transcriptional controls. Here we use mathematical modeling to investigate how transcriptional and post-transcriptional rhythms jointly control rhythmic gene expression. We particularly focus on rhythmic post-transcriptional regulation of the mRNA poly(A) tail, a nearly universal feature of mRNAs which controls mRNA stability and translation. Our model reveals that the rhythmicities in poly(A) tail length and mRNA translatability are most strongly affected by the rhythmicity in deadenylation, the process that shortens the poly(A) tail. Particularly, the phases of poly(A) tail length and mRNA translatability are dominated by the phase of deadenylation. In light of our findings, rhythmic control of deadenylation deserves greater future attention in the field of circadian gene expression.

## Introduction

Rhythmic control of gene expression is a hallmark of the circadian system. The daily rhythms in biochemistry, physiology and behavior ultimately stem from rhythmic gene expression in each cell [1, 2]. In mammals, approximately 3–15% of mRNAs are rhythmically expressed with a ~24 hr period in any given tissue [3–5]. The rhythmicity originates from a cell-autonomous circadian clock machinery, which consists of a set of core clock genes interlocked by transcription-translation feedback loops [6–8]. Many core clock genes encode transcription factors and interact with their respective target enhancers to exert rhythmic transcriptional control over mRNA expression [6, 9].

While rhythmic transcriptional control has been extensively studied, rhythmic control of gene expression also occurs beyond transcription [10–12]. Recent genome-wide analyses and mathematical modeling particularly highlight the role of post-transcriptional regulations in driving rhythmic mRNA expression [13–17]. Post-transcriptional regulations target various processes, such as splicing, nuclear export, cellular translocation, dormancy and degradation of RNAs [18]. Many post-transcriptional processes are under circadian control [10, 19–25]; these post-transcriptional processes, in turn, affect the phase and amplitude of the mRNA level. Ultimately, rhythmic transcription and post-transcriptional processes couple with each other and jointly determine the gene expression rhythm. For example, rhythmic RNA transcription and degradation jointly determine the rhythmicity in the mRNA level [16]. As yet, it remains unclear how the rhythmicities in other post-transcriptional processes affect the gene expression rhythm.

One of the post-transcriptional regulations that impact rhythmic gene expression is the regulation of poly(A) tail length. The tracts of adenosines at the 3' end of nearly all eukaryotic mRNAs are critical for controlling stability and translatability of the mRNAs [26–28]. Hundreds of mRNAs were discovered to exhibit robust circadian rhythms in their poly(A) tail lengths in mouse liver [29]. Interestingly, the rhythmicity in poly(A) tail length is closely correlated with the rhythmicity in the corresponding protein level, indicating that rhythmic poly(A) regulation plays an important role in driving rhythmic protein expression [29]. Similar daily fluctuations in poly(A) tail length also occur in mouse brain [30, 31]. In addition, the amplitude of mRNA rhythmicity increases in the absence of *Nocturnin*, a deadenylase (enzyme that removes poly(A) tails from mRNAs) which is rhythmically expressed in different mouse tissues [32, 33]. These observations underscore the importance of poly(A) tail rhythmicity in regulating circadian gene expression.

In this work, we built a mathematical model that describes mRNA dynamics under the regulation of rhythmic transcription, polyadenylation, deadenylation and degradation. We used the model to systematically examine how rhythmic expression and poly(A) tail regulation generates rhythmicities in poly(A) tail length and mRNA abundance. Our results highlight the rhythmicity in deadenylation as the strongest determinant for the rhythmicities in the poly(A) tail length and in the abundance of mRNAs with long poly(A) tails. The latter can be regarded as a rough proxy for mRNA translatability, because the poly(A) tail is known to regulate mRNA translation initiation [34–38]. Furthermore, deadenylase expression with several distinct peak phases, as those observed in the mouse liver [29], are able to override the impact from other rhythmic controls, and separate the peak phases of poly(A) tail length and abundance of long-tailed mRNAs into corresponding clusters. Finally, we used the model to predict factors or combination of factors (e.g., amplitudes of or phase differences between specific processes) that can explain the different classes of rhythmic characteristics found in mRNAs with rhythmic poly(A) tail length [29].

## Results

### Model for rhythmic mRNA and poly(A) tail regulation

In a typical RNA expression process, an RNA is first transcribed in the nucleus and acquires a long poly(A) tail as a result of nuclear polyadenylation [39]. After being exported into the cytoplasm, the mature mRNA undergoes deadenylation and is ultimately degraded [40]. Cytoplasmic polyadenylation, as another important post-transcriptional regulation, elongates the poly(A) tail to promote mRNA stability and translatability [41]. Although cytoplasmic polyadenylation is typically associated with translational control in oocyte maturation, early embryo development and synaptic plasticity [41–44], it is suggested to also play a role in circadian gene expression in mouse liver [29]. Furthermore, the expression level of *Gld2*, a poly(A) polymerase responsible for cytoplasmic polyadenylation, exhibits circadian rhythmicity in mouse liver [29]. In light of these biological facts, in the model we incorporated polyadenylation, together with transcription, deadenylation and degradation, to capture the major processes that dynamically regulate poly(A) tail length and mRNA abundance (**Fig 1A**). Note that the four processes can assume different amplitudes and phases for different genes, because these regulations can be mediated by different combinations of *cis*-elements and *trans*-factors [9, 13, 42, 45–47]. Instead of explicitly tracking the exact length of poly(A) tails, the model divides the mRNA population into a long-tailed fraction and a short-tailed fraction (**Fig 1A**), which mimics the fractionation conducted in the circadian transcriptome experiment (long-tailed >~60nt, short-tailed <~60nt, [29]). Herein we use the ratio between the abundances of long-tailed and short-tailed mRNAs as the metric for poly(A) tail length (**Fig 1B**), as was done in the experimental study [29].

For the sake of simplicity, we made the following assumptions in the model based on experimental evidence. First, degradation only occurs to the short-tailed mRNAs, because the poly(A) tail of an mRNA must be shortened to 10~15 nt before the mRNA is degraded [47–50]. Second, transcription and nuclear polyadenylation are lumped together, because transcription is followed by nuclear polyadenylation in general [51] and the poly(A) polymerases responsible for nuclear polyadenylation are not rhythmically expressed [29]. Taken together, in our model the transcription process directly leads to a long-tailed mRNA, the downstream cytoplasmic deadenylation and polyadenylation further mediate conversion between the long-tailed and short-tailed mRNAs, and degradation consumes the short-tailed mRNA (**Fig 1A**). The ordinary differential equations (ODEs) that govern the temporal dynamics of long-tailed (*L*) and

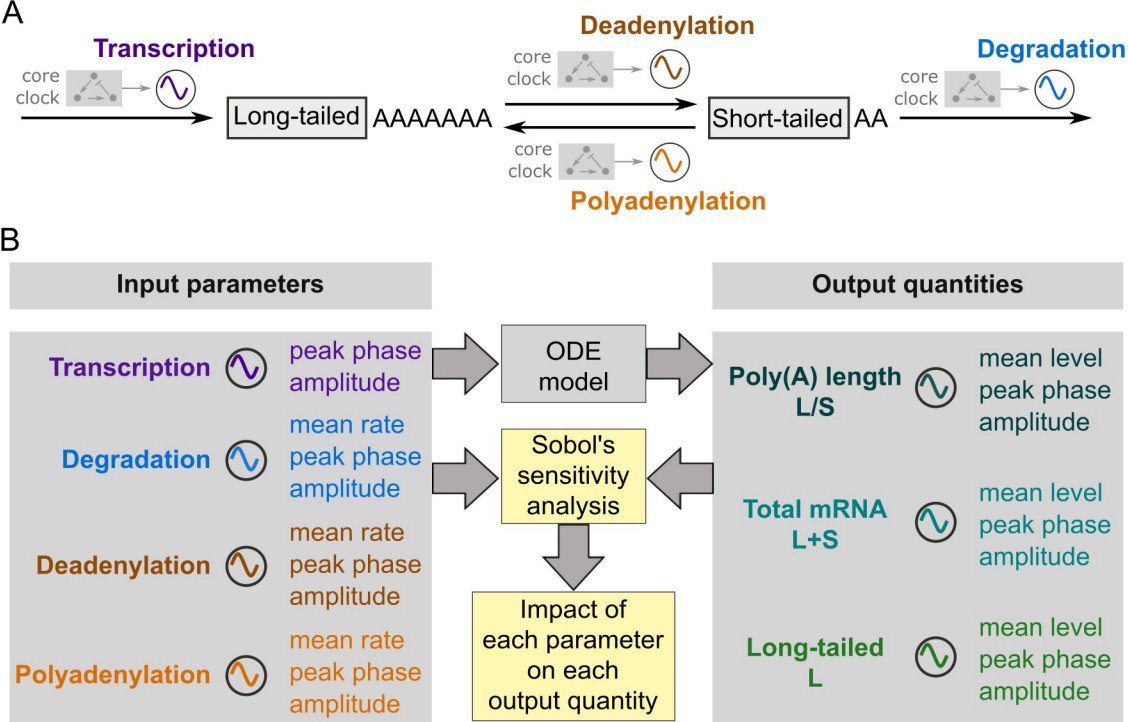

**Fig 1. Overview of the study.** (A) Schematic diagram of the model. The model describes four processes that control the poly(A) tail length and mRNA abundance: transcription, degradation, cytoplasmic deadenylation and polyadenylation. The rhythmicities of the four processes, i.e., amplitude and phase, are presumably controlled by the core clock mechanism (shaded molecular circuit), which is not explicitly included in the model. (B) Work flow of the study. Numeric simulations of the ODE model using different sets of input parameters (sampled according to **Table 1**, **S1 Fig**) generate the output quantities. The input parameters and output quantities are analyzed through the global parameter sensitivity analysis to quantify the impact of each parameter on each output quantity over the global parameter space.

short-tailed ($S$) mRNAs read as Eqs (1) and (2).

$$\text{Long}-\text{tailed mRNA}: \quad \frac{dL}{dt} = \underbrace{\kappa_{\text{trsc}}(t)}_{\text{transcription}} - \underbrace{\kappa_{\text{deA}}(t)L}_{\text{deadenylation}} + \underbrace{\kappa_{\text{polyA}}(t)S}_{\text{polyadenylation}} \tag{1}$$

$$\text{Short}-\text{tailed mRNA}: \quad \frac{dS}{dt} = \underbrace{\kappa_{\text{deA}}(t)L}_{\text{deadenylation}} - \underbrace{\kappa_{\text{polyA}}(t)S}_{\text{polyadenylation}} - \underbrace{\kappa_{\text{dgrd}}(t)S}_{\text{degradation}} \tag{2}$$

To capture the circadian rhythmicities of the four processes in Eqs (1) and (2), each reaction rate term $\kappa(t)$ is represented by a sinusoid function like Eq (3).

$$\kappa(t) = k(1 + A\cos(\omega(t - \varphi))) \tag{3}$$

where $k$ denotes the mean rate, $A$ the relative amplitude, and $\varphi$ the peak phase, of the process labeled by the subscript. The angular frequency, $\omega$, equals $2\pi/(24\text{hr})$. $\omega$ is fixed, while the other parameters vary. The subscript of a parameter indicates the process it describes (e.g., $k_{\text{deA}}$ stands for the mean deadenylation rate).

In this work we focus on how rhythmicities in the four processes affect the rhythmicities in total mRNA abundance and poly(A) tail length, because total mRNA abundance and poly(A)

tail length were quantified in the previous circadian transcriptome study [13, 29]. Additionally, we take the rhythmicity of long-tailed mRNA abundance as a rough proxy for the rhythmicity of mRNA translatability, because poly(A) tail facilitates translation initiation [34–38].

## Rhythmic deadenylation is the strongest contributor to rhythmicities in poly(A) tail length and long-tailed mRNA abundance

Because the parameters of the model are largely unknown and likely vary significantly from gene to gene, we need to investigate the dependency of the output rhythmicities on the input rhythmicities in the global parameter space (i.e., the entire possible range of parameter values). In the previous studies, such dependency has been analyzed by deriving approximate analytic solutions to models with up to two rhythmic input processes [16, 52]. With four rhythmic input processes in our model, the approximate analytic solution obtained using the same method as in [16, 52] are too complex to deliver any useful insight. We hence chose numeric simulations to investigate the input-output dependency for our model. We ran numeric simulations of the model (Eqs (1) and (2)) with random parameter values for the mean rates, relative amplitudes, and phases of each process (**Table 1** and **S1 Fig**). Only the mean rate of transcription was omitted, because it only affects the overall abundance of mRNAs, but not the output rhythmicity, i.e., the phases and relative amplitudes of mRNA abundance and poly(A) tail length (**S1 File**). From each simulated time trajectories $\{L(t), S(t)\}$, we extracted the peak phases, relative amplitudes and mean levels of total mRNA abundance ($L+S$), poly(A) length metric ($L/S$) and long-tailed mRNA abundance ($L$) (**Fig 1B**, also see Methods). These quantities were subject to further analysis, as elaborated in the following Results sections. For the rest of the paper, we will refer to these quantities, e.g., the peak phase of L/S ratio, generally as the "output quantities", unless any specific quantity is referred to.

Our model results reveal that the peak phase of deadenylation is the strongest contributor to the peak phase of L/S ratio (poly(A) length metric), followed by the peak phase of polyadenylation. Specifically, the scatter plot of the simulation results from random parameter sets demonstrates a strong dependency of the peak phase of L/S ratio on the peak phase of deadenylation, with a 10 ± 1.5 hr lag between the two (**Fig 2A**). The peak phase of L/S ratio also depends on the peak phase of polyadenylation, although much weaklier than its dependency on the peak phase of deadenylation (**Fig 2A**). In contrast, the peak phase of L/S ratio depends very little on the peak phases of transcription and degradation (**Fig 2A**).

To systematically quantify the impacts of each input parameter on each output quantity, we performed variance-based sensitivity analysis using the Sobol's method [53, 54] (**Fig 1B**, also

**Table 1. Parameter distribution for sampling.**

| Parameter | Symbol | Distribution | Source |
|---|---|---|---|
| Mean rate of transcription | $k_{\text{trsc}}$ | 1 (constant) | Has no effect on rhythmic patterns (**S1 File**) |
| Mean rate of degradation | $k_{\text{dgrd}}$ | $log_{10}(k/\text{hr}^{-1}) \sim \mathcal{N}(-1.10, 0.23^2)$ (Log-normal) | Fitting with half-life distribution measured in [72] |
| Mean rate of deadenylation | $k_{\text{deA}}$ | $log_{10}(k/\text{hr}^{-1}) \sim \mathcal{N}(-0.48, 0.23^2)$ (Log-normal) | Mean value of deadenylation rate estimated from [73]; deviation same as mRNA degradation |
| Mean rate of polyadenylation | $k_{\text{polyA}}$ | $log_{10}(k/\text{hr}^{-1}) \sim \mathcal{N}(-0.48, 0.23^2)$ (Log-normal) | Same as deadenylation |
| Relative amplitudes | $A_{\text{trsc}}, A_{\text{dgrd}}, A_{\text{deA}}, A_{\text{polyA}}$ | $A \sim U(0,1)$ (Uniform) | |
| Peak phases | $\varphi_{\text{trsc}}, \varphi_{\text{dgrd}}, \varphi_{\text{deA}}, \varphi_{\text{polyA}}$ | $\varphi \sim U(0,24)$ (Uniform) | |

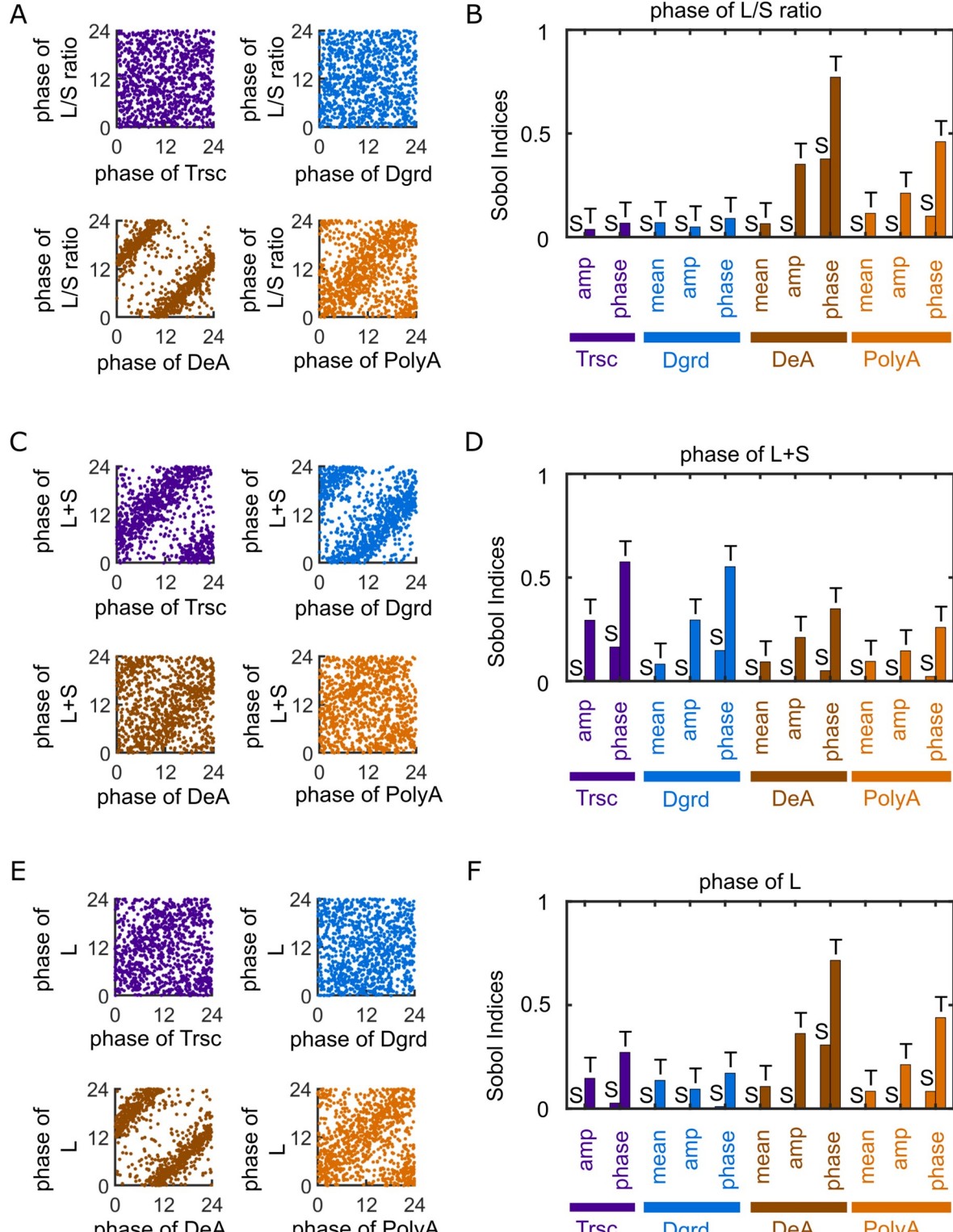

**Fig 2. Rhythmicities of poly(A) tail length and long-tailed mRNA abundance are strongly controlled by rhythmic deadenylation.** (A) Scatter plot of the peak phases of input processes versus the peak phases of L/S ratio (i.e., poly(A) length metric). (B) Sobol indices for the peak phase of L/S ratio. (C) Scatter plot of the peak phases of input processes versus the peak phases of L+S (i.e., total mRNA abundance). (D) Sobol indices for the peak phase of L+S. (E) Scatter plot of the peak phases of input processes versus the peak phases of L (i.e., long-tailed mRNA abundance). (F) Sobol indices for the peak phase of L. (A, C, E) Each scatter plot shows 10,000 data points randomly chosen

from the original simulations for the sake of visual clarity. (B, D, F) Bars with "S" on top: single Sobol indices. Bars with "T" on top: total Sobol indices. Mean values of the Sobol indices are shown, because the variances are too small for clear visualization (**S2 Fig**).

see Methods). Based on simulation results from a large number of random parameter sets spanning the global parameter space (**S1 Fig, Table 1**), the Sobol's method quantifies the sensitivity of an output quantity to an input parameter in terms of how much the parameter, due to the variation in its value, contributes to the variation in the output quantity. Specifically, the sensitivity is reported as the single (S) and total (T) Sobol indices, which represent the contribution of the parameter alone and the contribution of the parameter together with its (nonlinear) interactions with the other parameters, respectively (see Methods).

The estimated Sobol indices (**Fig 2B**) confirm the findings from the scatter plots (**Fig 2A**). For example, among all the input parameters, the peak phase of deadenylation has the largest Sobol indices with respect to the peak phase of L/S ratio. The values of the Sobol indices indicate that variance in the peak phase of deadenylation alone contributes to ~40% of variance in the peak phase of L/S ratio (longest "S" bar in **Fig 2B**). When the interactions of deadenylation with other processes are counted, this contribution increases to ~75% (longest "T" bar in **Fig 2B**). Additionally, the Sobol indices indicate that the relative amplitude of deadenylation has the strongest impact on the relative amplitude of L/S ratio (**S2 Fig**). In comparison, the mean level of L/S ratio, a quantity not related to rhythmicity, depends nearly equally on the mean rates of deadenylation and polyadenylation (**S2 Fig**). These results collectively demonstrate the rhythmicity in deadenylation as the strongest contributor to the rhythmicity in poly(A) tail length.

Our model results also show a significant impact of rhythmic deadenylation and polyadenylation on the rhythmicity of L+S (total mRNA abundance). Although the peak phases of transcription and degradation strongly influence the peak phase of L+S (**Fig 2C and 2D**), as expected, the Sobol indices indicate a weaker, yet substantial impact from the peak phases of deadenylation and polyadenylation on the peak phase of L+S (**Fig 2D**). These impacts can be understood from the regulation of mRNA stability by the poly(A) tail length, which is reflected in the model by the assumption that degradation is restricted to the short-tailed mRNAs (**Fig 1A**, Eqs (1) and (2)).

We further used the model to examine the effects of the four processes on the rhythmicity of mRNA translatability, using L (long-tailed mRNA abundance) as a proxy. Although L is a quantity directly related to both L+S level and L/S ratio, the Sobol indices show that the peak phase of L relies most heavily on the peak phase of deadenylation, followed by that of polyadenylation (**Fig 2F**). Consistently, the scatter plot shows a strong dependency of the peak phase of L on the peak phase of deadenylation, with an approximately 10 hr lag between the two (**Fig 2E**). This is a relationship highly similar to that observed between the peak phases of L/S ratio and deadenylation (**Fig 2A**). Furthermore, the relative amplitudes of deadenylation and polyadenylation are also among the strongest contributors to the relative amplitude of L (**S2 Fig**). Overall, the rhythmicities in deadenylation and polyadenylation make stronger impact on the rhythmicity of long-tailed mRNA abundance than the rhythmicities in transcription and degradation. This finding provides a possible explanation for the observed close correlation between the rhythmicities of poly(A) tail length and protein expression [29].

Cytoplasmic polyadenylation requires specific *cis*-elements in the 3' untranslated region (UTR) of an mRNA to recruit the molecular machinery that elongates the poly(A) tails [42]. However, such *cis*-elements do not necessarily exist in all mRNAs. Therefore, we also removed the polyadenylation term in our model and conducted the same global sensitivity analysis. The results demonstrate similar impacts of the rhythmicity of transcription, deadenylation and

degradation on the rhythmicity of L/S ratio, L+S and L (**S3 Fig**) as those found from the model with cytoplasmic polyadenylation (**Fig 2**, **S2 Fig**). Particularly, rhythmic deadenylation remains the strongest contributor to the rhythmicity of L/S ratio and L. Hence, our conclusion stays the same for mRNAs without the *cis*-elements that mediate cytoplasmic polyadenylation.

Taken together, these model results underscore the importance of rhythmic poly(A) regulation in circadian gene expression, especially its impact on the rhythmicity of poly(A) tail length, total mRNA abundance, and abundance of the long-tailed subpopulation. Importantly, deadenylation emerges as the strongest contributor to the rhythmicity of poly(A) tail length and long-tailed mRNA abundance.

## Rhythmic deadenylation can robustly cluster genes by their poly(A) tail rhythms

The rhythmicities in transcription, deadenylation, polyadenylation and degradation of mRNAs are ultimately controlled by the rhythmicities in the abundance and activity of the molecules mediating these processes, e.g., transcription factors, deadenylases and poly(A) polymerases. Interestingly, although the core clock machinery includes several transcription factors with different peak phases, the peak phases of nascent RNA synthesis (indicated by intron abundance) are strongly concentrated around ZT 15 (Zeitgeber time, where ZT 0 is defined as the time [hr] of lights on and ZT 12 is defined as the time of lights off) in mouse liver [13]. Additionally, a cytoplasmic poly(A) polymerase, *Gld2*, is rhythmically expressed with peak phase around ZT 3.5 [29]. Meanwhile, five deadenylases are also rhythmically expressed, with *Ccr4e/Angel1* peaking around ZT 2, *Ccr4a/Cnot6*, *Ccr4b/Cnot6l*, *Caf1a/Cnot7/ pop2* and *Parn* peaking around ZT 5, and *Ccr4c/Nocturnin* peaking around ZT 13 [29]. These data indicate that deadenylases assume a more diverse rhythmic expression pattern than poly (A) polymerases and nascent RNA transcription.

Intrigued by the above observation, we used our model to explore the potential consequence of having several distinct peak phases in deadenylases. In four separate *in silico* experiments, we set transcription, degradation, deadenylation or polyadenylation, respectively, to peak at three narrow windows centered around ZT 0, 8, and 16 (chosen to represent distinct time windows in general), while setting the peak phases of the other three processes to distribute evenly around the clock (**Fig 3ii–3v**). Our results demonstrate that, when deadenylation peaks in three narrow windows, the peak phases of L/S ratio and L are strongly clustered in three distinct windows (**Fig 3iv**). In contrast, when transcription (**Fig 3ii**), degradation (**Fig 3iii**) or polyadenylation (**Fig 3**v) peaks in three narrow windows, the resulting peak phases of L/S ratio and L do not show strong clustering. To test the effect of the actual rhythmic patterns observed in nascent RNA transcription and expression of deadenylases and polyadenylases, we set the distribution of peak phases centered around ZT 15 for transcription [13], narrow peak phase window centered around ZT 3.5 for polyadenylation, and narrow peak phase windows around ZT 2, ZT 5 and ZT 13 for deadenylation [29]. The simulation results demonstrate that the peak phases of both L/S ratio and L are strongly clustered into three distinct time windows (**Fig 3vi**). These results corroborate with the findings above about the strong impact of rhythmic deadenylation on the rhythmicities of L/S ratio and L (**Fig 2B and 2F**). Note that the mean rates and relative amplitudes of all four processes assumed random values in the model simulations (**Table 1**, **S1 Fig**). Therefore, our results indicate that multiple peak phases in deadenylation, but not other processes, can robustly cluster the peak phases of poly(A) tail length and mRNA translatability (~ long-tailed mRNA abundance) into distinct time windows, regardless of variations in the mean rates or rhythmicities of other processes.

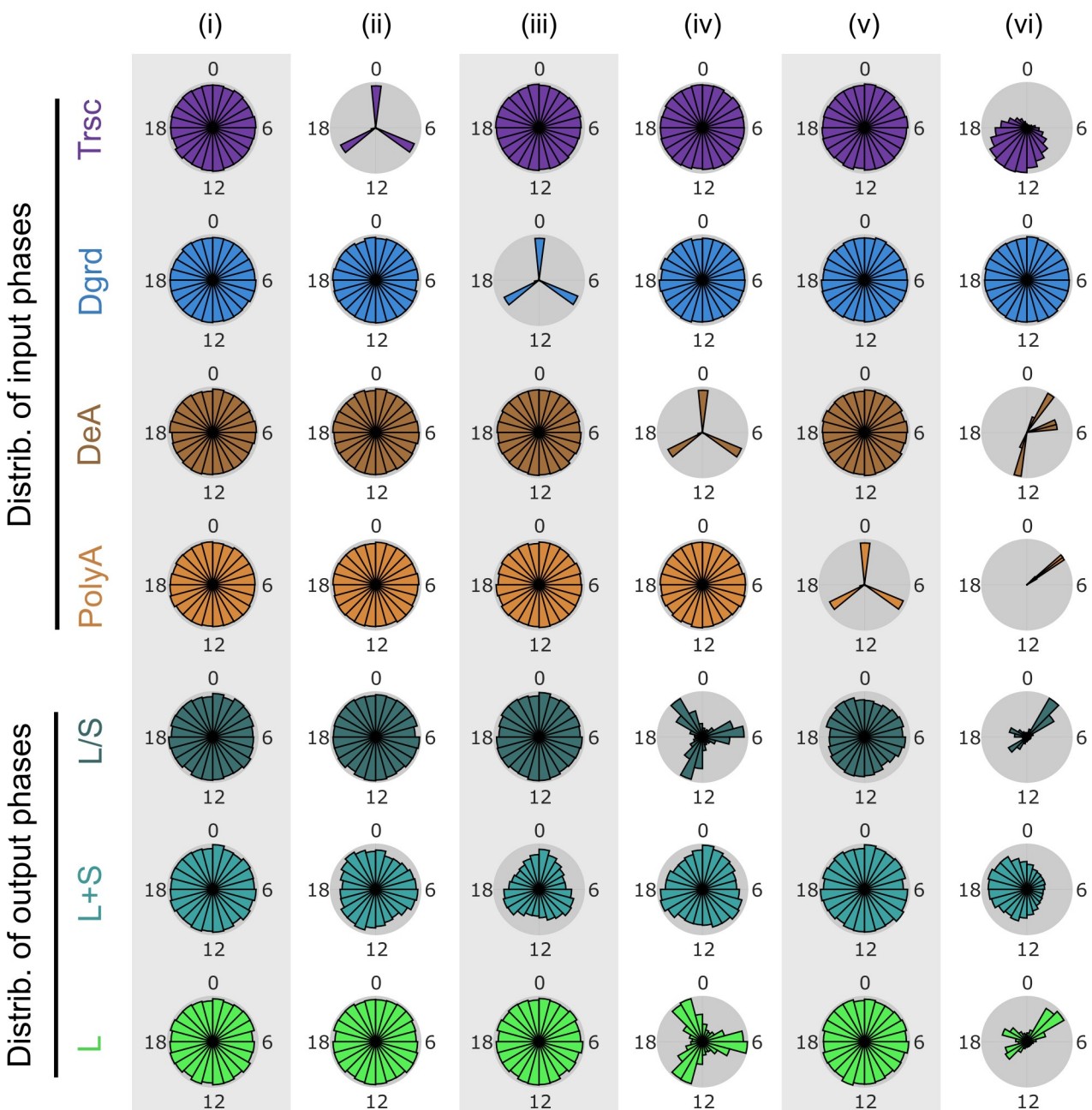

**Fig 3. Distinct peak phases in deadenylases cluster transcripts by their peak phases of poly(A) tail length and long-tailed mRNA abundance.** (i) Transcription, degradation, deadenylation and polyadenylation phases evenly distributed around the clock. (ii) Transcription phases within three narrow windows at ZT 0, 8, and 16. Degradation, deadenylation and polyadenylation phases evenly distributed around the clock. (iii) Degradation phases within three narrow windows at ZT 0, 8, and 16. Transcription, deadenylation and polyadenylation phases evenly distributed around the clock. (iv) Deadenylation phases within three narrow windows at ZT 0, 8, and 16. Transcription, degradation and polyadenylation phases evenly distributed around the clock. (v) Polyadenylation phases within three narrow windows at ZT 0, 8, and 16. Transcription, degradation and deadenylation phases evenly distributed around the clock. (vi) Peak phases of transcription follow transcriptome data reported by [13]. Deadenylation phases within three narrow windows at ZT 2, 5, and 13, and polyadenylation phases within one narrow windows at ZT 3.5, based on the data from [29], while degradation phases evenly distributed around the clock. Mean rates and relative amplitudes follow **Table 1** and **S1 Fig**.

## Factors that explain different classes of mRNAs with rhythmic poly(A) tail length

In the previous transcriptome-wide study [29], the mRNAs with poly(A) tail rhythmicity (PAR mRNAs) were grouped into three classes, based on their rhythmicities in pre-mRNA and total mRNA. The rhythmicity in pre-mRNA essentially reflects the rhythmicity in transcription. The Class I mRNAs are rhythmic not only in poly(A) tail length, but also in pre-mRNA and total mRNA (**Fig 4A**). The Class II mRNAs are rhythmic in poly(A) tail length and pre-mRNA, but not in total mRNA (**Fig 4A**). The Class III mRNAs are only rhythmic in poly(A) tail length, but not the other two (**Fig 5A**). Differences in mRNA half-lives were observed between the three classes and suggested to explain their differences in the rhythmic patterns of pre-mRNA, total mRNA, and poly(A) tail length [29]. Here we leverage our model to systematically identify factors that can lead to the combinatorial rhythmic patterns in these classes.

We first attempted to identify the model parameters that contribute most to the distinction between Class I and Class II. Because the only difference between Classes I and II is whether total mRNA abundance is rhythmic or not, we focused on identifying model parameters that contribute most to the relative amplitude of L+S. The Sobol indices reveal the mean degradation rate as the strongest contributor to the amplitude of L+S (**Fig 4B**). We then ran model simulations using random parameter sets (sampled from the distributions given in **Table 1** and **S1 Fig**) and identified the ones that exhibit the characteristics of Class I or Class II (**Fig 4A**). Out of all the random parameter sets, the mean degradation rates in the Class II parameter sets are overall lower than those in the Class I parameter sets (**Fig 4C**). This finding corroborates with the experimental observation that the average half-life (inversely proportional to the degradation rate) of Class II mRNAs is longer than that of Class I mRNAs [29].

The total Sobol indices also indicate that the peak phases of transcription and degradation as the second and third strongest contributors to the amplitude of L+S, respectively (**Fig 4B**). However, the corresponding single indices are diminishingly small (**Fig 4B**). The huge contrast between the total and single indices indicates that these two parameters exert strong impacts through interactions with other parameters. Because such huge contrasts between total and single indices do not exist in any other parameters, we speculated that the interactions likely happen between the two parameters themselves. Indeed, the Class I, but not the Class II, parameter sets, are strongly enriched with antiphasic rhythms between transcription and degradation (**Fig 4D**). This finding is consistent with the prediction by a previous modeling study that antiphasic coupling between rhythmic transcription and degradation enhances the rhythmicity of mRNA level [16].

The Sobol indices also reveal that the relative amplitudes of transcription and degradation and the mean deadenylation rate are potentially important contributors to the amplitude of L+S (**Fig 4B**). Indeed, the Class I parameter sets tend to have stronger amplitudes in transcription and degradation rates (**Fig 4E and 4F**), again, consistent with the previous modeling study [16]. Interestingly, unlike the Class I parameter sets (**Fig 4D–4F**, purple), the Class II parameter sets exhibit nearly even distributions of transcription-degradation phase difference, transcription amplitude and degradation amplitude (**Fig 4D–4F**, green). The distributions for Class I and Class II parameter sets indicate that generation of significant rhythmicity in L+S (Class I) requires sufficient phase difference between transcription and degradation, and sufficiently high amplitudes of transcription and degradation, simultaneously (**S4 Fig**). If any of these conditions are not satisfied, total mRNA abundance would not have significant rhythmicity (Class II). Lastly, the mean deadenylation rates in the Class I parameter sets tend to be larger than those in the Class II parameter sets (**Fig 4G**). This is related to the above finding about mRNA half-lives, because deadenylation promotes degradation and hence increasing

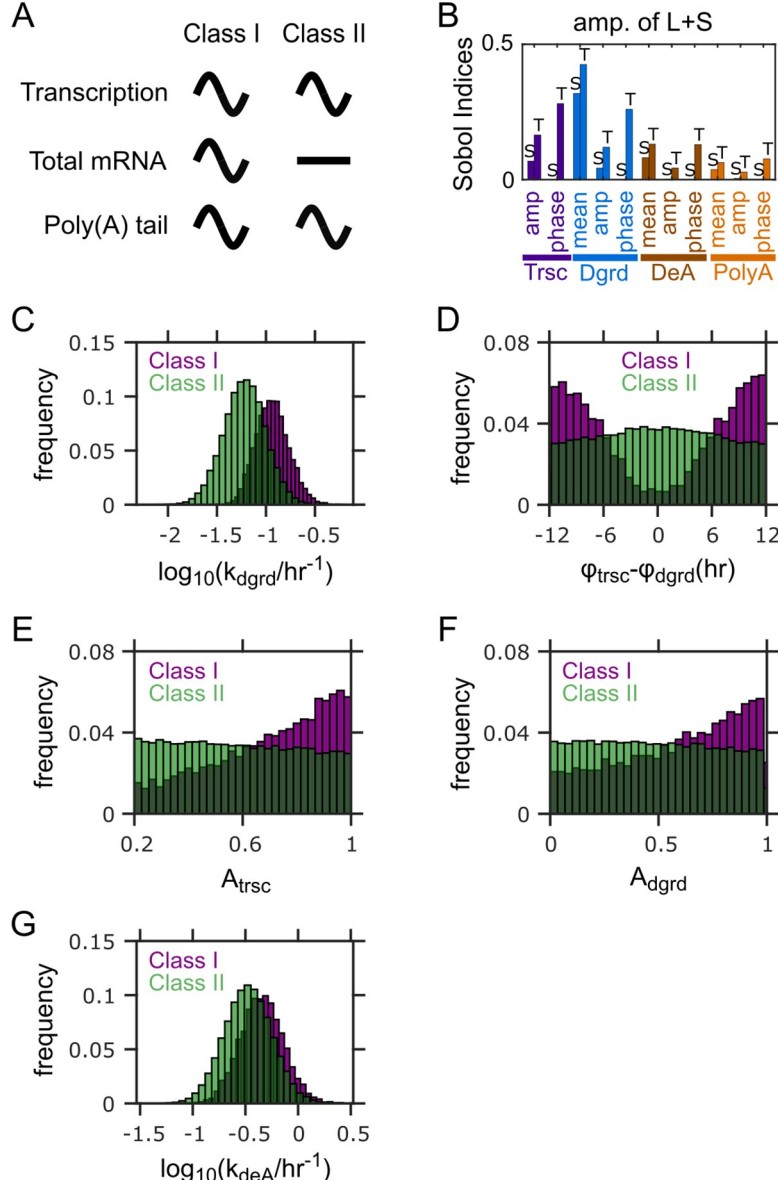

**Fig 4. Factors distinguishing between Class I and Class II PAR mRNAs.** (A) Characteristics of Class I and Class II PAR mRNAs. (B) Sobol indices for the amplitude of L+S (i.e., total mRNA abundance). Bars with "S" on top: single Sobol indices. Bars with "T" on top: total Sobol indices. (C) Distributions of mean mRNA degradation rates for the two classes. (D) Distributions of peak phase differences between transcription and degradation for the two classes. (E) Distributions of relative amplitudes of transcription for the two classes. (F) Distributions of relative amplitudes of degradation for the two classes. (G) Distribution of mean deadenylation rates for the two classes. Results in (C-G) from 100,000 simulations with parameters randomly sampled according to **Table 1**. Parameter sets with ≥0.2 relative amplitude in both L+S and L/S ratio are defined as Class I, while those with <0.2 relative amplitude in L+S and ≥0.2 relative amplitude in L/S ratio are defined as Class II.

the mean deadenylation rate has a similar effect on mRNA turnover as increasing the mean degradation rate.

Class III is distinct from Class I and Class II, since it does not have rhythmic transcription (**Fig 5A**). Because rhythmicity of transcription serves as an input to our model, we cannot use the model to identify the origin of lack of transcriptional rhythmicity. However, we are

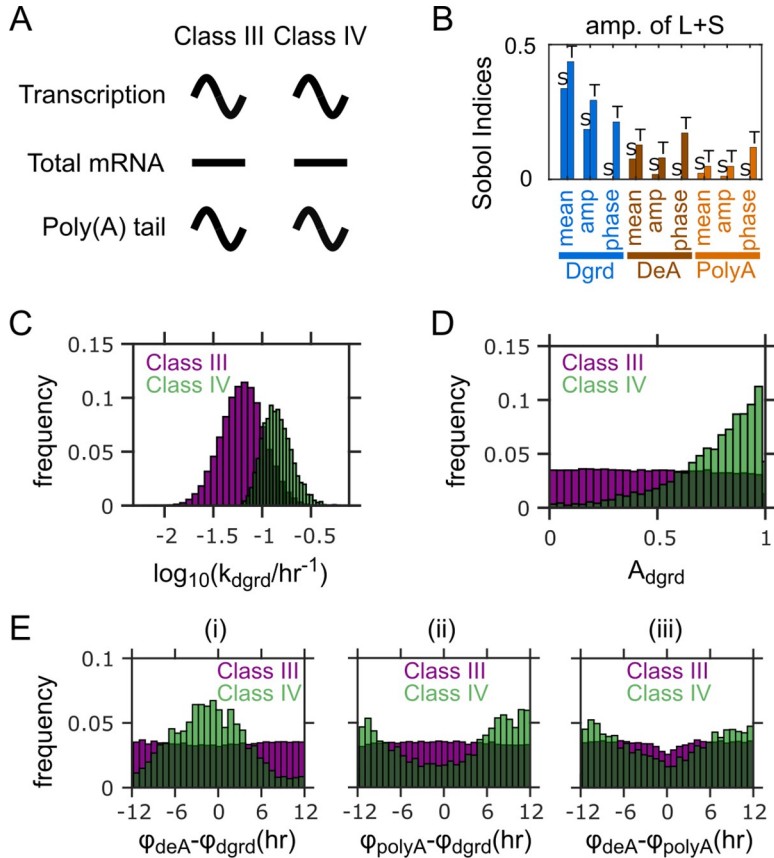

**Fig 5. Factors distinguishing between Class III and Class IV PAR mRNAs.** (A) Characteristics of Class III and the hypothetical Class IV mRNAs. (B) Sobol indices for the amplitude of L+S (i.e., total mRNA abundance) for the model without rhythmic transcription. Bars with "S" on top: single Sobol indices. Bars with "T" on top: total Sobol indices. (C) Distributions of mean mRNA degradation rates for the two classes. (D) Distributions of relative amplitudes of degradation for the two classes. (E) Distributions of peak phase differences (i) between deadenylation and degradation, (ii) between polyadenylation and degradation, and (iii) between deadenylation and polyadenylation for the two classes. Results in (C-E) from 100,000 simulations with parameters randomly sampled according to **Table 1**, but without rhythmic transcription ($A_{trsc} = 0$). Parameter sets with ≥0.2 relative amplitude in L/S ratio and <0.2 relative amplitude in L+S are defined as Class III, while those with and ≥0.2 relative amplitude in both L/S ratio and L+S are defined as Class IV.

interested in understanding why all PAR mRNAs without transcriptional rhythmicity also lack rhythmicity in L+S [29]. For the convenience of discussion, we use "Class IV" to refer to a hypothetical group of PAR mRNAs that would exhibit rhythmicity in total mRNAs and poly (A) tails, but not in pre-mRNA (**Fig 5A**); this group of mRNAs are not found in the experiments [29]. We used the model to identify model parameters that could contribute to the difference between Class III and the hypothetical Class IV. Because both Class III and Class IV do not have rhythmic transcription, we ran model simulations with non-rhythmic transcription (i.e., setting the relative amplitude of transcription to zero, while keeping the other parameters sampled from the same distributions as before (**Table 1**, **S1 Fig**)). Out of the random parameter sets, we identified those that fit the characteristics of Class III or Class IV (**Fig 5A**). We also calculated the Sobol indices for this model.

When the model does not have rhythmic transcription, the Sobol indices again reveal the mean degradation rate as the strongest contributor to the relative amplitude of L+S (**Fig 5B**). Consistently, the Class IV parameter sets require much larger mRNA degradation rate, i.e.,

much shorter mRNA half-life, than the Class III parameter sets, to sustain rhythmic total mRNA (**Fig 5C**). Therefore, the absence of Class IV mRNAs from the experimental observations is most likely due to the long half-lives of the mRNAs without rhythmic transcription. Indeed, Class III has the longest average mRNA half-life measured among all mRNAs that are rhythmically expressed [29].

We also identified a few additional factors that could distinguish Class III from Class IV. Based on the Sobol indices, the second strongest factor affecting the amplitude of L+S is the relative amplitude of degradation (**Fig 5B**). The Class IV parameter sets have markedly higher amplitudes of degradation than Class III (**Fig 5D**). The phases of all three rhythmic processes, i.e., degradation, deadenylation and polyadenylation, are also potentially important contributors, because their total Sobol indices are substantial (**Fig 5B**). Again, the huge contrast between the total and single indices for these phase parameters, but not the other parameters, suggests that they exert impacts through interactions among themselves. We hence examined the distribution of pairwise differences between the three phase parameters. The Class IV parameter sets are significantly enriched in the region where the peak phases of deadenylation and degradation are close to each other, but opposite to that of polyadenylation (**Fig 5E**). This can be understood from the fact that both deadenylation and degradation promote mRNA turnover while polyadenylation inhibits it. Unlike the Class IV parameter sets, no distinct patterns are found in the amplitude of degradation or the phase differences in the Class III parameter sets (**Fig 5D and 5E**). Similar to the discussion above for Class I and Class II, these results indicate that the Class IV characteristics require both sufficiently large amplitude in degradation and sufficient differences of the polyadenylation phase from the deadenylation and degradation phases (**S5 Fig**). The missing of Class IV from the experiment suggests that mRNAs without transcriptional rhythmicity may also fail to satisfy these conditions at the same time.

Overall, our model suggests that besides mRNA half-life, relative amplitudes and phase difference between transcriptional and post-transcriptional processes can also contribute to the rhythmic characteristics that distinguishe the three observed classes of PAR mRNAs (**Figs 4 and 5**). These results highlight that rhythmic transcriptional and post-transcriptional processes collectively determine the rhythmicity in mRNA expression and poly(A) tail length. It will be of future interests to test if the factors predicted by the model are indeed correlated with different rhythmic characteristics.

## Discussion

In this work, we developed a parsimonious mathematical model (**Fig 1**) to quantitatively evaluate how rhythmic inputs from transcription, degradation, polyadenylation and deadenylation collectively determine the rhythmic outputs in mRNA abundance, poly(A) tail length and mRNA translatability (~long-tailed mRNA abundance). Our model results and global sensitivity analyses reveal rhythmic deadenylation as the strongest factor in controlling the peak phases and amplitudes of rhythmic poly(A) tail length and long-tailed mRNA abundance (**Figs 2 and 3**). Our model also suggests how three classes of rhythmic characteristics observed in PAR mRNAs [29] arise from the dynamic features of the four processes, as well as the coupling among their rhythmicities (**Figs 4 and 5**).

Many post-transcriptional steps are involved in regulating circadian gene expression [10, 11]. The importance of dynamic coupling between rhythmic transcription and post-transcriptional processes was demonstrated by a previous modeling study by Lück et al. [16]. That work particularly highlights that rhythmic turnover is necessary for achieving >6 hr peak phase difference between transcription and mRNA abundance. In comparison, our study explicitly considers the effects of poly(A) regulation, a common intermediate process in the mRNA decay

pathway, on rhythmic gene expression. In our model, the dynamic coupling among rhythmic transcription, polyadenylation, deadenylation and degradation determines the rhythmic patterns in both poly(A) tail length and mRNA abundance. These four processes jointly regulate the circadian gene expression driven by the core clock (**Fig 1A**), with a principle similar to a previous theoretical study that investigates rhythmic fluxes along metabolic chains using circadian response analysis [52]. Because deadenylation is necessary for mRNA degradation and polyadenylation opposes it, rhythmic deadenylation and polyadenylation, unsurprisingly, affect the rhythmicity of total mRNA abundance at a level comparable to rhythmic degradation (**Fig 2**, **S2 Fig**). However, when poly(A) tail length and its effect on mRNA translatability are considered, rhythmic deadenylation emerges as the most important rhythmic controller (**Fig 2, S2 Fig**). This finding highlights the crucial role of rhythmic poly(A) regulation in circadian gene expression. Of course, our model has not included other mRNA decay pathways that do not depend on poly(A) regulation, such as endonuclease cleavage of mRNA followed by 5'-3' decay [47]. For any mRNA decayed through these pathways, which are less common, their expression rhythms obviously would not depend on the rhythmicity in poly(A) regulation.

Based on the finding of rhythmic deadenylation as the strongest contributor to rhythmicity of poly(A) tail length and long-tailed mRNA abundance, we further discovered that rhythmic deadenylation is capable of synchronizing the target circadian gene expression post-transcriptionally. According to the model results, three distinct peak phases in deadenylation activity, as those suggested in mouse liver [29], can robustly cluster the mRNAs into three distinct groups by their peak phases of poly(A) tail length and long-tailed mRNA abundance; this deadenylation rhythm-dependent clustering happens regardless of the rhythmicity in the other processes (**Fig 3**). This finding suggests a potential mechanism to synchronize the expression of genes controlled by the same deadenylases, which would foster synergy among these genes around the clock. This synchronization potential is unique to rhythmic deadenylation, but not the other rhythmic processes (**Fig 3**).

The potential capability of deadenylation to synchronize circadian gene expression further poses two interesting questions. First, could deadenylation help synchronize circadian gene expression among different cells and entrain their cell-autonomous clocks to the systemic rhythms? Recent studies suggest that rhythmic feeding or other systemic rhythmic cues control the rhythmic expression of several deadenylases, including *Parn*, *Pan2* [17] and *Nocturnin* [55], through clock-independent pathways. Given our findings, such systemically driven rhythmicity in deadenylases could dictate the rhythmicity of poly(A) tail length and mRNA translatability (~long-tailed mRNA abundance). This could help synchronize circadian gene expression in cells influenced by the same systemic signals. Second, could deadenylases play a role in tissue-specific circadian gene expression? Rhythmic gene expression is known to vary tremendously from tissue to tissue: different tissues not only share very few rhythmically expressed genes beyond the core clock genes, but also display different peak times for some genes [5, 56, 57]. It is puzzling how the rhythmicity in gene expression varies so much across different tissues while the cellular clock machineries are the same and are presumably synchronized throughout the organism. Most previous studies on the mechanisms of tissue-specific circadian gene expression have focused on tissue-specific transcriptional control, such as rhythmic fluctuations in chromatin structure and interactions between core clock transcription factors and tissue-specific transcription factors [58, 59]. In light of the findings from our work, differential expression patterns of deadenylases in different tissues [60] could serve as an additional mechanism to mediate tissue-specific circadian gene expression. These two interesting questions await future studies to answer.

In our current model, the poly(A) regulation has been coarse-grained as one-step conversions between a long-tailed and a short-tailed mRNA subpopulations. Such coarse-graining

retains the most essential kinetic features of the poly(A) regulation processes, while allowing for significant reduction of the model and efficient global parameter sensitivity analysis. From such analysis we identified the critical role of deadenylation in rhythmic regulation. In reality, both deadenylation and polyadenylation act sequentially, i.e., adding or subtracting one adenosine at a time. Unlike one-step chemical reactions, the kinetics of sequential processes are often non-exponential [61, 62]. To evaluate the rhythmicities of poly(A) regulation and gene expression more accurately, we will include a linear reaction chain in the model to account for sequential steps of deadenylation and polyadenylation in our future work.

Circadian gene expression is a critical, yet highly complex process. Expressing the right genes at the right time and the right place requires coordinated control at various gene expression steps, as well as across different cells and tissues. Systems-level study of the coupling between different rhythmic processes is necessary to gain comprehensive understanding of circadian gene expression control, and more importantly, the ability to make positive use of circadian rhythm in disease treatments. As our work demonstrates the significant impact of rhythmic poly(A) regulation and its coupling with rhythmic mRNA transcription and degradation on circadian gene expression, it will be of great future interest to examine how coupling of rhythmicities in all transcriptional, post-transcriptional, translational and post-translational processes influences circadian gene expression.

Finally, the methodology used in this study, namely, global parameter sensitivity analysis over randomized model parameters, are broadly applicable to modeling studies in chronobiology. Randomized global parameter sweeping is effective and efficient for model analysis, when the model parameters are largely unknown or highly variant (e.g., high variations across different genes for parameters in our model), and the corresponding experimental data are too sparse to effectively constrain the parameters. Results from global parameter sweep provide insights about which elements of the system are important for the target qualitative or quantitative behaviors. Many chronobiology models fall in this type. In fact, similar randomized global parameter sweep was used to identify components that are critical to generate key characteristis of the circadian clock, such as circadian entrainment, adaptation to seasonal changes in photoperiod, and tissue-specific rhythms [63–65].

In addition, the Sobol's method serves as a particularly powerful tool for parameter sensitivity analysis for models in chronobiology. In chronobiology models, oscillation phases are often important quantities of interest. As circular variables, i.e., ZT 0 = ZT 24, phases are intrinsically nonlinear and non-monotonic. Analyzing nonlinear and non-monotonic variables using classic correlation and dependency analyses, such as Pearson correlation and Spearman correlation, could lead to misleading conclusions, because these methods are based on assumptions about linear and monotonic relations between the analyzed data. For example, in our model, Pearson and Spearman correlation analyses demonstrate strong negative correlation between the phases of deadenylation and L/S ratio, a spurious conclusion due to the circular nature of phases (**S6 Fig**); other pairs of input and output phases suffer different levels of distortion in their Pearson and Spearman correlations (**S6 Fig**). Based on variance decomposition (see Methods), the Sobol's method circumvents these problems and can effectively analyze nonlinear and non-monotonic variables [54]. The method can be used widely in chronobiology models to identify the key factors that drive phases of target quantities, such as the phase difference between PER2 and TP53, whose interaction is critical for the crosstalk between the circadian clock and cell cycle [66]. Furthermore, the Sobol's method would be useful in model-driven chronopharmacology research [67–69], a particularly exciting new area, to elucidate the molecular mechanism of the therapy or drug and the source of variations in the therapeutic effect.

Control of circadian rhythm is a great example of systems biology topics, since the circadian control is intricately connected to many, if not all, biological processes from the cellular to organismal levels. Like research on other systems biology topics, combination between computational modeling and experimentation provides a powerful tool and will accelerate future advance in the research of circadian control.

## Methods

### Model simulation and extraction of phase, amplitude and mean from simulation results

For any given parameter set, Eqs (1) and (2) were simulated using the ODE solver, ode45, in MATLAB. For a simulated time trajectory $\{L(t), S(t)\}$, the peak phases, relative amplitudes and mean levels of $L(t)+S(t), L(t)/S(t)$ and $L(t)$ were analyzed. First, the time trajectory for the output quantity of interest, e.g., $L(t)/S(t)$, was calculated from $\{L(t), S(t)\}$. Then a 48-hr window after 700 hrs (sufficiently long to pass the initial transient) was extracted from the trajectory for data analysis. The trajectories typically have irregular time spacing (due to automatic time stepping in the ode45 solver) and hence often have insufficient time resolution for accurate determination of the peak phase. To make accurate estimation of the peak phase, the 48-hr trajectory was interpolated upon 500 equally spaced time points spanning the 48 hrs. The peak phase was evaluated from the time for the maximum interpolated value, $t_{max}$, i.e., peak phase = $mod(t_{max}+700,24)$ (hr). The mean value was estimated by taking the average of the interpolated values. The relative amplitude was estimated by taking the maximum and minimum interpolated values and calculating (max−min)/(2×mean). An output quantity was considered rhythmic if its relative amplitude is equal to or greater than 0.2.

### Parameter sampling

We performed global parameter sensitivity analysis [70] on the model to analyze the general contribution of each parameter to each output quantity (i.e., peak phase, relative amplitude and mean of $L(t)+S(t), L(t)/S(t)$ and $L(t)$). In this study we drew random parameter values from the distributions listed in **Table 1** and plotted in **S1 Fig**. The peak phases and relative amplitudes were sampled from uniform distributions of their possible ranges by definition (**Table 1**). The mean reaction rates were sampled from log-normal distributions suggested by previous genomic scale measurements (see sources indicated in **Table 1**). We set the mean transcription rate as constant, as it only causes proportional changes in $L(t)$ and $S(t)$, and does not affect the rhythmic patterns of any quantity (**S1 File**). To improve the accuracy of the global sensitivity analysis for models with many parameters, one needs parameter samples that well represent the parameter space. To this end, we used the sampling method of Latin hypercube [71], which is known to ensure good representation of a high-dimensional parameter space.

### Sobol's method of global sensitivity analysis

To evaluate the impact of each model parameter (e.g., phase of deadenylation) on each model output (e.g., relative amplitude of L/S ratio), we used a variance-based global parameter sensitivity analysis method, the Sobol indices [53, 54]. The conceptual basis of this method is functional decomposition of the variance of an output $Y$ into contributions from each parameter and interactions between the parameters (Eq (4)).

$$\text{Var}(Y) = \sum_i V_i(Y) + \sum_{i<j} V_{ij}(Y) + \cdots + V_{1,2,\ldots,k}(Y) \qquad (4)$$

In Eq (4), $V_i(Y) = \mathrm{Var}_{X_i}(\mathrm{E}_{X_{\sim i}}(Y|X_i))$ is the contribution from the $i$-th parameter alone. Here $X_{\sim i}$ denotes the combined parameter set except for the $i$-th parameter. $\mathrm{E}_{X_{\sim i}}(Y|X_i)$ denotes the expectation of output $Y$ conditional on a fixed value for $X_i$ (while the other parameters randomly vary). $\mathrm{Var}_{X_i}(\mathrm{E}_{X_{\sim i}}(Y|X_i))$ then denotes the variance of the calculated conditional expectation as $X_i$ varies. The second term of Eq (4), $V_{ij}(Y) = \mathrm{Var}_{X_i,X_j}(\mathrm{E}_{X_{\sim ij}}(Y|X_i,X_j)) - V_i(Y) - V_j(Y)$, is the contribution from the interactions between the $i$-th and $j$-th parameters, where $X_{\sim ij}$ denotes the combined parameter set except for the $i$-th and $j$-th parameters. Contributions from higher-order interactions between parameters are defined similarly as $V_{ij}(Y)$.

The Sobol indices are then defined as fractions of the decomposed terms in Eq (4) out of the total variance, $\mathrm{Var}(Y)$. In practice, only the single (Eq (5)) and total-effect (Eq (6)) indices are calculated because relatively simple algorithm as described below can be designed. Specifically, the single Sobol index, $S_i$, characterizes the contribution of variance in $X_i$ alone to the total variance in $Y$ (Eq (5)). The total-effect, or total index, $S_{Ti}$, characterizes the contribution of variance in $X_i$, as well as the variance caused by its coupling with other parameters, to the total variance in $Y$ (Eq (6)). Conveniently, the total-effect contribution equals $\mathrm{E}_{X_{\sim i}}(\mathrm{Var}_{X_i}(Y|X_{\sim i}))$. Here $\mathrm{Var}_{X_i}(Y|X_{\sim i})$ denotes the variance of output $Y$ conditional on a fixed set of $X_{\sim i}$ (while $X_i$ randomly varies). $\mathrm{E}_{X_{\sim i}}(\mathrm{Var}_{X_i}(Y|X_{\sim i}))$ then denotes the expectation of the calculated variance as $X_{\sim i}$ varies (Eq (6)). The larger the single and total indices are, the more sensitive $Y$ is to $X_i$, or the more impact $X_i$ has on $Y$.

$$S_i = \frac{\mathrm{Var}_{X_i}(\mathrm{E}_{X_{\sim i}}(Y|X_i))}{\mathrm{Var}(Y)} \tag{5}$$

$$S_{Ti} = \frac{V_i + \sum_{j \neq i} V_{ij} + \sum_{j \neq i, k \neq i, j < k} V_{ijk} + \cdots}{\mathrm{Var}(Y)} = \frac{\mathrm{E}_{X_{\sim i}}(\mathrm{Var}_{X_i}(Y|X_{\sim i}))}{\mathrm{Var}(Y)} \tag{6}$$

We followed the specific algorithms given in [53] and [74] for evaluating the single (Eq (5)) and total indices (Eq (6)). The details of implementation are explained below.

1. Sample from the distributions given in **Table 1** two independent groups of $N$ parameter sets ($N = 100{,}000$ in this study):

$$A = \begin{bmatrix} A_{1,1} & \cdots & A_{1,k} \\ \vdots & \ddots & \vdots \\ A_{N,1} & \cdots & A_{N,k} \end{bmatrix}, B = \begin{bmatrix} B_{1,1} & \cdots & B_{1,k} \\ \vdots & \ddots & \vdots \\ B_{N,1} & \cdots & B_{N,k} \end{bmatrix} \tag{7}$$

Each row in $A$ and $B$ represents one set of $k$ parameters. $k = 11$ for the model with cytoplasmic polyadenylation. $k = 8$ for the model without cytoplasmic polyadenylation. $k = 9$ for the model without transcriptional rhythmicity.

2. Construct $k$ hybrid groups of parameter sets. The $i$-th hybrid group, $A_B^{(i)}$, has the $i$-th column equal to the $i$-th column of $B$, and the remaining columns copied from $A$, where

$i = 1,\ldots,k.$

$$A_B^{(i)} = \begin{bmatrix} A_{1,1} & \cdots & B_{1,i} & \cdots & A_{1,k} \\ A_{2,1} & \cdots & B_{2,i} & \cdots & A_{2,k} \\ \vdots & \ddots & \vdots & \ddots & \vdots \\ A_{N,1} & \cdots & B_{N,i} & \cdots & A_{N,k} \end{bmatrix} \tag{8}$$

3. Estimate the total variance for each model output, $Y_q$.

$$\mathrm{Var}(Y_q) \approx \frac{1}{2N} \sum_{n=1}^{N} \{ [f_q(A_{(n)}) - f_q(\bar{A}_{(n)})]^2 + [f_q(B_{(n)}) - f_q(\bar{B}_{(n)})]^2 \} \tag{9}$$

where $f_q$ denotes the $q$-th output quantity (**Fig 1B**) from the circadian gene expression model (Eqs (1) and (2)). $A_{(n)}$ and $B_{(n)}$ denote the $n$-th parameter set (row) in Groups $A$ and $B$, respectively. The bars on top denote the average of output quantities over $N$ parameter sets.

4. For each pair of parameter $X_i$ and output $Y_q$ in the model (**Fig 1B**), estimate the single and total Sobol indices, using Eqs (10) and (11) [53, 74].

$$S_{iq} \approx \frac{1}{N} \sum_{n=1}^{N} f_q(B_{(n)}) [f_q((A_B^{(i)})_{(n)}) - f_q(A_{(n)})] / \mathrm{Var}(Y_q) \tag{10}$$

$$S_{Tiq} \approx \frac{1}{2N} \sum_{n=1}^{N} [f_q((A_B^{(i)})_{(n)}) - f_q(A_{(n)})]^2 / \mathrm{Var}(Y_q) \tag{11}$$

where $(A_B^{(i)})_{(n)}$ denotes the $n$-th parameter set (row) in the $i$-th hybrid group, and the other notations follow those described above.

## Supporting information

**S1 Fig. Sampling distributions of the model parameters.** (A) Sampling distribution of mean mRNA degradation rate. (B) Sampling distribution of mean deadenylation rate. (C) Sampling distribution of mean polyadenylation rate. (D) Sampling distribution of relative amplitudes of all rhythmic processes. (E) Sampling distribution of peak phases of all rhythmic processes.
(TIF)

**S2 Fig. Sobol indices of the model with cytoplasmic polyadenylation.** Calculation using Eqs (1) and (2). Label "S" on top: single Sobol indices. Label "T" on top: total Sobol indices. Error bars show the standard deviation of the estimated Sobol indices from 10 repeats. Each repeat was performed using the procedure described in Methods with $N = 100,000$.
(TIF)

**S3 Fig. Sobol indices of the model without cytoplasmic polyadenylation.** Calculation using Eqs (1) and (2), with $k$polyA = 0. Label "S" on top: single Sobol indices. Label "T" on top: total Sobol indices. Error bars show the standard deviation of the estimated Sobol indices from 10 repeats. Each repeat was performed using the procedure described in Methods with $N = 100,000$.
(TIF)

**S4 Fig. Two-parameter distributions show a more confined distribution of the Class I parameter sets than the Class II sets.** (A) Parameter distributions with respect to the amplitudes of transcription and degradation. (B) Parameter distributions with respect to the phase difference between transcription and degradation and the amplitude of transcription. (C) Parameter distributions with respect to the phase difference between transcription and degradation and the amplitude of degradation. Case (i): Scatter plots for 3,000 Class I sets and 3,000 Class II sets randomly chosen from the 100,000 parameter sets used to produce Fig 4. Case (ii): The parameter sets in case (i) that satisfy $-1.15 \leq \log_{10} k_{\mathrm{dgrd}} \leq 0$. Case (iii): The parameter sets in case (i) that satisfy $-2 \leq \log_{10} k_{\mathrm{dgrd}} \leq -1.15$. As the mean degradation rate, $k_{\mathrm{dgrd}}$, decreases, fewer Class I parameter sets are found in a more confined region.
(TIF)

**S5 Fig. Two-parameter distributions show a more confined distribution of the Class IV parameter sets than the Class III sets.** (A-C) Parameter distributions with respect to the degradation amplitude and the phase difference between deadenylation and degradation (A), or between polyadenylation and degradation (B), or between deadenylation and polyadenylation (C). (D-F) Parameter distributions with respect to pairs of phase differences. Case (i): Scatter plots for 3,000 Class III sets and 3,000 Class IV sets randomly chosen from the 100,000 parameter sets used to produce Fig 5. Case (ii): The parameter sets in case (i) that satisfy $-1 \leq \log_{10} k_{\mathrm{dgrd}} \leq 1$. Case (iii): The parameter sets in case (i) that satisfy $-1.5 \leq \log_{10} k_{\mathrm{dgrd}} \leq -1$. As the mean degradation rate, $k_{\mathrm{dgrd}}$, decreases, fewer Class IV parameter sets are found in a more confined region.
(TIF)

**S6 Fig. Comparison among Pearson correlation, Spearman correlation and Sobol indices.** (A) Dependency analyses between the phase of L/S ratio and the phase of each input. (B) Dependency analyses between the phase of L+S and the phase of each input. (C) Dependency analyses between the phase of L and the phase of each input. Scatter plots from Fig 2 for each input-output pair are placed below the corresponding dependency analysis results.
(TIF)

**S1 File. Setting mean transcription rate as constant does not affect rhythmic pattern.**
(PDF)

## Acknowledgments

We thank Dr. Xi Chen (Virginia Tech) for helpful discussion of the Sobol method.

## Author Contributions

**Conceptualization:** Shihoko Kojima, Jing Chen.

**Data curation:** Xiangyu Yao.

**Formal analysis:** Xiangyu Yao.

**Investigation:** Xiangyu Yao, Shihoko Kojima, Jing Chen.

**Methodology:** Xiangyu Yao, Jing Chen.

**Project administration:** Jing Chen.

**Resources:** Jing Chen.

**Software:** Xiangyu Yao.

**Supervision:** Jing Chen.

**Validation:** Xiangyu Yao, Jing Chen.

**Visualization:** Xiangyu Yao, Jing Chen.

**Writing – original draft:** Xiangyu Yao, Shihoko Kojima, Jing Chen.

**Writing – review & editing:** Xiangyu Yao, Shihoko Kojima, Jing Chen.

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
