## [Decision Letter · Decision Letter 0]

6 Feb 2020

Dear Dr. Chen,

Thank you very much for submitting your manuscript "Critical role of deadenylation in regulating poly(A) rhythms and circadian gene expression" for consideration at PLOS Computational Biology.

As with all papers reviewed by the journal, your manuscript was reviewed by members of the editorial board and by several independent reviewers. In light of the reviews (below this email), we would like to invite the resubmission of a significantly-revised version that takes into account the reviewers' comments.

We cannot make any decision about publication until we have seen the revised manuscript and your response to the reviewers' comments. Your revised manuscript is also likely to be sent to reviewers for further evaluation.

Sincerely,

Attila Csikász-Nagy

Associate Editor

PLOS Computational Biology

Jason Haugh

Deputy Editor

PLOS Computational Biology

Reviewer's Responses to Questions

**Comments to the Authors:**

Reviewer #1: In this manuscript Yao et al. used mathematical modeling to explore the possible roles of deadenylation in regulating circadian gene expression. I read this work with great interest. It has been an active research area on how circadian rhythm is regulated, and this work shows there is still new mechanism that has not been thoroughly investigated. Through analyzing simple mathematical models that are based on existing experimental observations, these authors convincingly showed that Poly(A) tail length regulation is an important part of circadian gene regulation. I only have a few minor comments for the authors to consider.

1) It took me some efforts to realize that the authors focused on the downstream genes that are regulated by the circadian rhythm network. The authors can make this point more transparent by modifying Fig. 1A. Specifically they can add schematically the circadian rhythm oscillator as an upstream network that affects the four processes, transcription, deadenylation, etc.

2) Figure 3 is probably the most interesting result of this work. I notice the phase plots and some polar coordinate numbers overlap in the figures. The authors may adjust the plotting.

3) It is impressive that some rich information can already deduced from the model given by equations 1 and 2. It is justified to start with such simple models, which are convenient for thorough parameter space examination. There are, however, certain limitations on coarse-graining the poly-A tail length as two discrete states. Mathematically one can treat the process as a reaction-diffusion process, and can write down either a discrete chemical master equation or continuous 1D process (described by Langevin equations or Fokker-Planck equations). The authors may discuss limitation of their formulation and alternative future studies.

Reviewer #2: Attached

Reviewer #3: The authors address a hot topic – the differential regulation of clock-controlled genes (CCGs). It has been found earlier (Naef, Westermark, Robles, Koike, Heyd …) that different stages of gene products display differential rhythmicity. Kojima et al. 2012 provided a rich resource of rhythmic genes associated with mRNA polyadenylation. In the submitted manuscript a theoretical framework is provided to connect different processes to measured quantities.

The paper is clearly written and the selected model type is consistent with the sparse experimental data. Technically speaking, linear non-autonomous differential equations are simulated for 11 randomly selected parameters. Scatter plots and global sensitivity analysis help to quantify the dominant regulations. The theoretical findings are connected to observations (phases of CCGs, phases of deadenylases). Below I list some suggestion to improve the already very good manuscript.

Minor comments:

1. I am aware that redundancy is helpful but abstract, summary, introduction, discussion are quite redundant. Practically identical sentences should be avoided. In the Author Summary a different level of abstraction compared to the abstract might help readers.

2. Regarding posttranscriptional circadian regulations some more references might be of interest: F. Heyd regarding splicing, M. Young, M. Brunner and K. Vanselow regarding nuclear import/export).

3. K. Thurley (PNAS 2017) developed a general theory of rhythmic fluxes along chains and introduced circadian response analysis (CRA). These results are highly relevant to interpret the model in Figure 1.

4. Page 5: …recently …. 2012???

5. Linear systems as in Equations 1-3 can be treated analytically using e.g. variation of constants (see also Luck & Westermark and Thurley 2017). I am aware that the resulting expressions with 4 periodic terms become very complicated. Anyway, the authors should discuss why they prefer numerical solutions.

6. I appreciate the analysis of random parameters since few quantitative details are available. This approach of studying parameter sets instead of specific parameters became popular in chronobiology and the authors might cite papers using this approach: J. Locke BMC 2008, P. Pett LSA 2018, B. Ananthasubramaniam JMB 2020).

7. Regarding Figure 2 I am surprised that DeA controls L/S and L strongly but S only weakly. Typically, an oscillating denominator (S in case of L/S) has a strong impact. Is there any explanation why L seems more relevant regarding DeA regulation.

8. Looking at Figure 1, I am not surprised that Deadenylation plays a major role. It has a central position and modulates the decay of L and the production of S. If this is the correct interpretation I would expect for constant DeA a major control of Polyadenylation on the L/S ratio since this reaction is also at the center of the chain and effects L and S.

**Have all data underlying the figures and results presented in the manuscript been provided?**

Reviewer #1: Yes

Reviewer #2: Yes

Reviewer #3: Yes

PLOS authors have the option to publish the peer review history of their article (what does this mean?). If published, this will include your full peer review and any attached files.

Reviewer #1: No

Reviewer #2: No

Reviewer #3: No
---

## [Decision Letter · Decision Letter 1]

2 Apr 2020

Dear Dr. Chen,

We are pleased to inform you that your manuscript 'Critical role of deadenylation in regulating poly(A) rhythms and circadian gene expression' has been provisionally accepted for publication in PLOS Computational Biology.

Best regards,

Attila Csikász-Nagy

Associate Editor

PLOS Computational Biology

Jason Haugh

Deputy Editor

PLOS Computational Biology

Reviewer's Responses to Questions

**Comments to the Authors:**

Reviewer #1: The authors have satisfactorily addressed my concerns.

Reviewer #2: The authors have fully answered to comments.

Reviewer #3: The authors addressed carefully all comments. Congratulations to such an interesting combination of data, insight, modeling, and statistics.

**Have all data underlying the figures and results presented in the manuscript been provided?**

Reviewer #1: None

Reviewer #2: Yes

Reviewer #3: None

PLOS authors have the option to publish the peer review history of their article (what does this mean?). If published, this will include your full peer review and any attached files.

Reviewer #1: No

Reviewer #2: No

Reviewer #3: No

---

## [Editor Report · Acceptance letter]

20 Apr 2020

PCOMPBIOL-D-20-00037R1 

Critical role of deadenylation in regulating poly(A) rhythms and circadian gene expression

Dear Dr Chen,

I am pleased to inform you that your manuscript has been formally accepted for publication in PLOS Computational Biology. Your manuscript is now with our production department and you will be notified of the publication date in due course.

With kind regards,

Laura Mallard
